# Manuka Honey Induces Apoptosis of Epithelial Cancer Cells through Aquaporin-3 and Calcium Signaling

**DOI:** 10.3390/life10110256

**Published:** 2020-10-27

**Authors:** Simona Martinotti, Giorgia Pellavio, Mauro Patrone, Umberto Laforenza, Elia Ranzato

**Affiliations:** 1DiSIT- Dipartimento di Scienze e Innovazione Tecnologica, University of Piemonte Orientale, viale Teresa Michel 11, 15121 Alessandria, Italy; simona.martinotti@uniupo.it (S.M.); mauro.patrone@uniupo.it (M.P.); 2DiSIT- Dipartimento di Scienze e Innovazione Tecnologica, University of Piemonte Orientale, piazza Sant’Eusebio 5, 13100 Vercelli, Italy; 3Department of Molecular Medicine, Human Physiology Unit, University of Pavia, 27100 Pavia, Italy; giorgia.pellavio01@universitadipavia.it (G.P.); lumberto@unipv.it (U.L.)

**Keywords:** AQP3, Ca^2+^ signaling, honey, manuka, ROS

## Abstract

Honey is a natural product with a long use in traditional medicine and is well recognized to regulate different biological events. It is an important source of various biological or pharmacological molecules and, therefore, there is a strong interest to explore their properties. Evidence is growing that honey may have the potential to be an anticancer agent acting through several mechanisms. Here we observed for the first time in a cancer cell line a possible mechanism through which honey could induce an alteration in the intracellular reactive oxygen species and homeostatic balance of intracellular calcium concentration leading to cell death by apoptosis. This mechanism seems to be enhanced by manuka honey’s ability to maintain high H_2_O_2_ permeability through aquaporin-3.

## 1. Introduction

Honey, achieved from nectar collected by honey bees, is a combination of carbohydrates, proteins, fatty acids, minerals and vitamins containing several classes of phytochemicals with a high flavonoid content and the presence of phenolic compounds [1]. 

Honey has been utilized for a long time as a traditional remedy and one of the ancient known utilizations is for the healing of wounds. Honey antibacterial activity has been well defined in the literature and some intrinsic characteristics of honey such as acidity and high osmolarity as well as the occurrence of flavonoids and phenolic acids are recognized as important for these activities [1].

Moreover, there is a growing number of widespread scientific and clinical indications to suggest honey use for wound healing and tissue repair [1,2,3].

In addition to its antibacterial and wound promoting abilities, recent data have underlined multiple roles for honey in inflammatory cytokines release by macrophages [4], neutrophil migration stimulation [5], cell proliferation inhibition and apoptosis induction as well as the arrest of cell cycle [6] and lipoprotein oxidation inhibition [7].

More recently, research has revealed that honey, with its richness in flavonoids and polyphenols, shows antiproliferative properties against tumor cell lines [6,7,8].

Nevertheless, its antitumor mechanisms are still to be completely explained. A few pathways through which natural honey could produce its antitumor properties have been proposed such as permeabilization of the mitochondrial outer membrane, arrest of the cell cycle, apoptosis induction and oxidative stress modulation [9].

Aquaporins are transmembrane proteins originally recognized as water channels in all organisms and then found to show multiple substrate specificity, such as hydrogen peroxide (H_2_O_2_) [10,11].

We have already demonstrated that honey is able to produce H_2_O_2_ [2] and in keratinocytes, a specific aquaporin (i.e., aquaporin-3) helps the passive H_2_O_2_ diffusion across the biological membranes [2]. The H_2_O_2_ mediated transport through aquaporin-3 (AQP3) is of physiological prominence for downstream cellular signaling pathways such as the intracellular Ca^2+^ signals onset [12,13].

Here, we describe a mechanism through which manuka honey induces an alteration in the intracellular ROS and homeostatic balance of [Ca^2+^]_i_ leading to cell death by apoptosis in a cancer cell line. This mechanism is enhanced by manuka honey’s ability to maintain high H_2_O_2_ permeability through AQP3.

These results raise our knowledge and could be advantageous for honey application as a therapeutic candidate for targeting tumor cells.

## 2. Materials and Methods

### 2.1. Honey Sample

Honey specimens of diverse floral origin, i.e., manuka (UMF, Unique Manuka Factor, 15+, 250 mg/kg methylglyoxal, 250+ MGO) buckwheat and acacia less than 12 months old were received from Yamada Apiculture Center, Inc. (Tomata-gun, Okayama, Japan). Raw honeys were maintained at room temperature in the dark. A stock honey solution was arranged by dissolving in a warmed DMEM (Dulbecco’s Modified Eagle’s medium) or loading buffer and 0.22 µm sterilized. Honey preparations were freshly made before each experiment.

### 2.2. Cell Culture and Reagents

All reagents, if not specified, were bought from Mercks/Sigma-Aldrich.

A431 cells (derived from an epidermal carcinoma of the vulva taken from an 85 year old female) were maintained at 5% CO_2_, 37 °C, in 4.5 g/L (high glucose) DMEM complemented with streptomycin (100 mg/mL), 10% FBS, penicillin (100 U/mL) and L-glutamine (200 mM) (FBS, Euroclone, Milan, Italy) [14].

### 2.3. Calcein-Am Assay

The cell viability assay was executed by using calcein-acetoxymethylester (Calcein-AM), a nonfluorescent, lipophilic dye. Calcein-AM enters the cells and is converted to hydrophilic fluorescent dye in the cytoplasm by intracellular esterases. A431 cells seeded in 96-well plates were exposed for 24 h to honey as specified, then PBS-washed and maintained at 37 °C for half an hour with a probe (Calcein-Am prepared in PBS, 2.5 µM). Fluorescence values were then obtained with a multimode reader (Infinite 200 Pro, Tecan, Wien, Austria) with the use of a 485 nm excitation filter and a 535 nm emission filter.

### 2.4. Apoptosis Assay

Apoptosis induction with honey in A431 cells was assessed by a multi-parameter apoptosis assay kit (catalog #600330, Cayman Chemicals Company, Ann Arbor, MI, USA). Cells seeded in 96-well plates were exposed for 3 h to honey as specified and analyzed using a multimode reader (Infinite 200 Pro, Tecan).

### 2.5. Free Cytosolic Ca^2+^ Concentration ([Ca^2+^]_i_) Measurements

Cells, seeded and settled down overnight on glass-based dishes (Iwaki Glass, Inc., Tokyo, Japan), were loaded in the dark at 37 °C for 30 min with 20 mM Fluo-3/AM, a fluorescent, cell-permeant calcium probe. A loading buffer consisting of (mM) 1 MgCl_2_, 5 KCl, 10 glucose, 2 CaCl_2_, 140 NaCl and 10 ph 7.4 HEPES was utilized. To perform experiments avoiding the presence of Ca^2+^ (0 Ca^2+^ condition), the ion was absent from the confocal buffer [14,15,16,17]. Cells were then analyzed with a time-lapse setting, utilizing a confocal apparatus (Zeiss LSM 510 system) equipped with an inverted microscope (Carl Zeiss Microscopy GmbH, Jena, Germany).

Excitation was produced by an argon source (488 nm) and the emission was gathered by a broad bandpass filter. To reduce the Fluo-3 bleaching, the laser power was reduced to 1%. Cells were observed with a 20× Zeiss objective (0.5 NA). Fluo-3 fluorescence was measured using the ROI-mean tool of the Zeiss software.

The calibration of Fluo-3 probe was realized using this approach [18]:Ca^2+^ = Kd(F − Fmin)/(Fmax − F)
where Kd is 400 nmol/L.

Fmax and Fmin are respectively the maximum and minimum of fluorescence levels gained by Fluo-3 after 500 μM A23187 (calcium ionophore) exposure followed by a 20 mM EDTA addition.

### 2.6. Polymerase Chain Reaction (PCR)

After cell exposure to the designated experiment settings, a commercial kit was used to collect and purify total RNA (NucleoSpin RNAII Kit from Macherey-Nagel, Düren, Germany). cDNA was created with a specific cDNA kit (from Roche Diagnostics, Transcriptor First Strand cDNA Synthesis Kit). Quantitative reverse transcriptase PCR (qRT-PCR) was realized using a Sybr green mastermix (Ambion Inc, Austin, TX, USA) and a panel of primers (KiCqStart^®^ SYBR^®^ Green Primers; Table 1) by means of a PCR machine (CFX384 Real-Time machine from Bio-Rad Laboratories, Hercules, CA, USA). A ∆∆Ct method was utilized to calculate the gene expression.

### 2.7. Immunoblotting

A431 cell cultures were homogenized using a RIPA buffer additioned with a phosphatase and protease inhibition mixture. Homogenates were treated at 80 °C for 10 min in a Laemmli buffer [19]. A total of 30 µg proteins were electrophorized on a precast polyacrylamide gel (4–20% Mini-PROTEAN TGX Stain-Free Gels, Bio-Rad Laboratories) and a PVDF membrane was blotted by using a Trans-Blot Turbo Transfer Pack (Bio-Rad Laboratories) with a specific transfer system (Bio-Rad Laboratories).

To prevent non-specific protein binding, PVDF membranes were blocked using a blocking solution consisting of Tris buffered saline solution (TBS) prepared with 5% skimmed milk and 0.1% Tween.

Membranes were then probed for one hour or overnight with a rabbit anti-AQP3 antibody (SAB5200111 with a dilution of 1:1000) and a RabMAb anti-beta-2-microglobulin antibody (EP2978Y) Abcam, product number: ab75853 with a dilution of 1:10,000) diluted in the TBS and 0.1% Tween. After washing, the membranes were exposed for at least 1 h with a goat antirabbit secondary antibody conjugated with peroxidase (AP132P; Millipore, with a dilution of 1:100,000) prepared in a blocking solution. The bands were visualized by incubating with western blotting revealing apparatus (CYANAGEN, Italy). To approximate the band molecular weights, pre-stained molecular weight markers (ab116028, Abcam) were utilized. Blots were scanned with an Expression 1680 Pro scanner system (Epson Corp., Long Beach, CA, USA). The bands were quantified by densitometry (Amersham) and the outcomes indicated as a densitometric ratio of AQP3/B2M.

### 2.8. RNA Interference

The N-ter Nanoparticle approach was used to transfect cells with 5 μM siRNA oligonucleotides, or with equimolar scrambled siRNA. We utilized commercial siRNA sequences specific to the human AQP3 (see Table 2). Commercial non-targeting siRNA (siRNA Universal Negative Control) was utilized for scrambled siRNA experiments. Transfected cells were harvested after 24 h and utilized for the designated tests.

### 2.9. Intracellular ROS Measurement

The level of ROS present in the cells was evaluated utilizing dihydrorhodamine (DHR)-123, a fluorescent dye precursor, transformed to fluorescent rhodamine 123 upon interaction with ROS. Cells seeded in 96-well plates were loaded at room temperature in the dark for half an hour with DHR-123 (30 μM) in a loading buffer as described for the confocal microscopy experiment. After incubation, cells were washed with a loading buffer and the fluorescence was measured with a multimode reader (Infinite 200 Pro, Tecan) by using a 485 nm excitation filter and a 530 nm emission filter. ROS production observations were indicated as arbitrary units of fluorescence [20].

### 2.10. Water Permeability Measures

Osmotic water permeability of A431 cells was assessed by a stopped-flow light scattering method as already defined [21].

Water transport was assessed in (a) control, untreated cells, (b) cells exposed to 50 µM H_2_O_2_ for 45 min and (c) cells treated with 4% manuka honey for 45 min. The iso-osmolarity of control and H_2_O_2_-treated cells was obtained by adding 4% artificial honey [2] (3 g saccharose, 11.17 g glucose, 13.5 g fructose and 5.7 mL water) to the incubation media.

### 2.11. Statistical Analysis

Statistics were made with GraphPad Prism 8 (GraphPad Software Inc, San Diego, CA, USA). Based on the data, one-way or two-way ANOVAs were utilized and the required corrections (Tukey’s test, Bonferroni correction, Dunnet post-test and Newman–Keuls Q test) were subsequently applied. Statistical details of each experiment (test used, value of n, replicates, p value, etc.) can be found in the Figure legends.

## 3. Results

### 3.1. Cell Viability

We evaluated honey cytotoxicity by using a Calcein-Am end-point on A431 cells, an epidermoid carcinoma cell line. We used three different kind of honeys such as acacia, buckwheat and manuka, as shown in Table 3.

According to the cell viability assay results, we performed the subsequent experiments only with manuka honey, which was the most cytotoxic one.

### 3.2. Intracellular Ca^2+^ Variations

We started to examine if manuka honey was capable of producing alterations in [Ca^2+^]_i_. Accordingly, we measured variations in intracellular Ca^2+^ induced after honey treatment by means of time-lapse confocal microscopy imaging of A431 cells preloaded with Fluo-3/AM, a fluorescent Ca^2+^ probe.

We observed that the [Ca^2+^]_i_ sampled at 5 s intervals (Figure 1A) and at 0.5 s intervals (Figure 2A) did not experience in control conditions any spontaneous oscillations.

We analyzed [Ca^2+^]_i_ variations of A431 cells exposed to a range of increasing concentrations (1, 2, 3, 4 and 5% v/v) of manuka honey, observing a dose-dependent increase in the [Ca^2+^]_i,_ (Figure 1).

The recorded traces showed that from 1 to 3%, manuka honey induced a non-significant peak that returned to a plateau phase comparable with control conditions (Figure 1A,B). Conversely, 4 and 5% manuka honey determined a consistent [Ca^2+^]_i_ peak that was not able to return to a homeostatic plateau phase (Figure 1A,B). In particular, as shown in Figure 1C, in these conditions, the peak and the plateau phases were not significantly different, indicating an altered calcium homeostasis.

Considering the incompatibility with cell survival of the treatment with 5% manuka honey, despite the short observation time we decided to perform the following Ca^2+^ signals recording only with 4% manuka honey.

Moreover, to highlight the importance of the presence of honey for the maintenance of the altered plateau phase, we performed an observation treating cells with manuka honey after 60 s as in the other experiments. Immediately after reaching the peak phase, we then removed the honey treatment, replacing this with only a loading buffer. We observed that the absence of honey in the medium determined a significant decrease in the [Ca^2+^]_i_ that reached control values immediately after the removal of honey (Figure 2B).

### 3.3. Origin of Ca^2+^ and Ca^2+^ Toolkit Involvement

We repeated the previous experiment in a 0 Ca^2+^ condition (i.e., the absence of Ca^2+^ from the extracellular space) and the result showed the disappearance of the Ca^2+^ peak after 4% manuka honey exposure but a slightly increase in the plateau phase. (Figure 3A). To remark the need for the presence of calcium in the extracellular space, we performed an observation treating cells with manuka honey. Immediately after the peak phase, we then maintained honey treatment but in a 0 Ca^2+^ condition. We noticed that the absence of extracellular calcium determined a gradual decrease in the [Ca^2+^]_i_ reaching control values (Figure 3B).

These results highlighted that extracellular Ca^2+^ entry showed a crucial role in the [Ca^2+^]_i_ rise in A431 cells treated with manuka honey.

To further evaluate the involvement of Ca^2+^ entry from extracellular space, we performed the experiment in the presence of 4% manuka honey and econazole, a TRPM2 inhibitor (10 µM, 30 min pre-incubation). Confocal imaging showed that the inhibitor presence was able to abrogate the Ca^2+^ peak after 4% manuka honey exposure but we recorded in the second part of the observation a subsequent small calcium increase as previously detected in a 0 Ca^2+^ condition (Figure 4A).

Furthermore, we utilized two inhibitors of the IP_3_ signaling pathway, i.e., U73122, a PLC inhibitor [14] (10 µM, 30 min pre-incubation) and caffeine, a blocker of IP_3_R [15] (10 mM, 30 min pre-incubation). In both conditions we observed a decrease of about 2.5 time of the Ca^2+^ peak (Figure 4B).

### 3.4. Apoptosis Induction

Based on cell viability results and calcium homeostasis alteration observations, we assessed the induction of apoptosis evaluating with a TMRE probe the mitochondrial membrane potential (ΔψM). After manuka honey treatment, we observed a significant reduction of ΔψM highlighting a strong dose-dependent induction of cell death (Figure 5A). This result was also confirmed with the evaluation of the Annexin V positivity (Figure 5B). As the reduction of ΔψM is known to produce an increase in the ROS level in the cytosolic milieu, we also evaluated by means of the DHR-123 probe the variation of intracellular ROS after treatment with 4% manuka honey. As in Figure 5C, after 45 min of exposure we detected a doubling of the fluorescence value.

### 3.5. ROS Involvement in the Mechanism of Action of Manuka Honey

It is known that honey induces the production of H_2_O_2_ in the extracellular medium [1] and that the extracellular presence of this species is fundamental for the biological honey mechanism of action [2]. Furthermore, we repeated the confocal observation of A431 cells under 4% manuka honey in the presence of catalase (CAT, 500U). We observed that CAT acts as “scavenger” for free radicals produced by honey and therefore drastically abrogates the [Ca^2+^]_i_ rise (Figure 6A). Accordingly, H_2_O_2_ is the most suitable candidate to induce Ca^2+^ signaling after manuka honey exposure.

Starting from this result, we performed again the cytotoxicity assay treating cells with an increasing range of manuka honey concentrations after CAT pretreatment. The EC_50_ value was 10.23% (confidence interval (CI) 8.84–11.83%).

Moreover, to understand if extracellular H_2_O_2_ is fundamental in the apoptosis induction, we evaluated again the variation of ΔψM after manuka honey treatment in presence or not of CAT. We observed a significant reduction of ΔψM only in the absence of CAT while honey plus CAT did not determine any significant variation with respect to untreated cells, highlighting the pivotal role played by ROS in the induction of apoptosis. This result was also validated evaluating the Annexin V positivity (Figure 6B).

### 3.6. Role of Aquaporins (AQPs) in Honey Toxicity

We have already demonstrated that aquaporins (AQPs) are able to mediate the passage of H_2_O_2_ from extracellular space to cytosol during honey and propolis exposure [2,22]. In particular, we have demonstrated the role of AQP3 in intracellular ROS level increases.

To this aim, we have quantified the basal expression of some AQPs in A431 cells and their variations upon manuka honey exposure (Figure 7). For these experiments, we utilized a honey concentration of 2%, which resulted in the highest not toxic concentration based on confocal calcium recordings. Only the AQP3 expression was improved after honey treatment.

To evaluate the mechanism supported by AQP3 in manuka honey induced toxicity, we accomplished the Calcein-Am assay after silencing by RNAi of AQP3 (Figure 8A,B). The results showed that EC_50_ for scrambled cells was 3.56% (CI 2.58–4.37%) while for AQP3-siRNA cells it was 9.19% (CI 7.16–11.8%), endorsing the pivotal role of AQP3 in mediating cytotoxicity in A431 cells.

Moreover, we assessed if AQP3 was the mediator of the H_2_O_2_ entry able to increase intracellular ROS. By using a stopped-flow light scattering technique, we determined the osmotic water permeability of A431 cells, which is an index to the H_2_O_2_ permeability. The results show that the presence of 50 M H_2_O_2_ (a concentration similar to that obtained by honey treatment) reduced significantly the AQP permeability while, with manuka honey, the permeability was unaltered with respect to the control condition. These data support the involvement of AQP3 in the entrance of H_2_O_2_ into the cells and that one or more substances present in the manuka honey were able to maintain the pore completely open even with high concentrations of H_2_O_2_ (Figure 8C,D). Furthermore, we evaluated, by means of a DHR-123 probe, the generation of intracellular ROS after the addition of the 4% manuka honey that was abrogated in the presence of AQP3 RNAi (Figure 8E). Likewise, after honey exposure, the rapid increase in [Ca^2+^]_i_ was completely erased upon RNAi silencing of AQP3 (Figure 8F).

## 4. Discussion

A tumor is one of the most shared reasons of death and is a significant health burden [20] and the number of new tumor cases per year is estimated to rise. Neoplastic condition is still a challenge even with the growing research on its prevention and cure. During the last years, classic methods for cancer treatment have showed severe negative effects. Therefore, researchers were fascinated towards less toxic approaches and novel procedures.

There is, therefore, an augmented awareness for complementary and alternative medicine practice for a huge number of conditions from acute to chronic and deadly diseases [23]. Furthermore, there has been better attention on chemo-preventive and chemo-therapeutic agents derived from food or natural products [24]. The relative safety of food-derived compounds [25] makes them a very interesting and alternative approach compared with classic tumor therapies.

Honeys among natural products are the most investigated for their possible antitumor properties [23]. A few authors have highlighted that honey may support the basis for the growth of novel therapeutics for patients with tumor and tumor-related conditions. Jungle honeys showed the induction of chemotaxis for neutrophils and ROS production, demonstrating its anticancer activity [5]. Recent works on some human tumor cells such as cervical, breast, oral and osteosarcoma [26,27,28] using Malaysian jungle honey displayed significant antitumor activity. Honey has also showed to possess anticancer properties in vivo and in vitro in an experimental bladder model [28].

Honey contains a huge amount of phytochemicals such as high flavonoid and phenolic content, which contribute to its action [1]. Honey contains sugars, proteins, organic acids, vitamins, phenolic and volatile compounds. The chemical volatile composition of honey is of great importance for influencing its organoleptic properties. Among these volatile compounds we can consider aldehydes, alcohols, esters, ketones, benzene derivatives, nitrogen containing compounds and carboxylic acids. Currently, more than several hundred volatile molecules have been recognized in honeys of diverse botanical origins [23,25].

A few differences in honey efficacy are due its various floral sources as well as floral sources may possess different active molecules. Manuka honey has recently gained attention for its biological activities particularly for its antioxidant and antibacterial capacities. A few observations sustain manuka honey utilization in skin regenerative medicine [1,2,29]. Another interesting component of manuka honey is methylglyoxal (MGO). This compound, normally formed during the Maillard reaction, has been recognized as an important contributor to the non-peroxide antibacterial activity of manuka honey [1].

Buckwheat honey shows a characteristic dark color and its antioxidant ability is well known [1]. Buckwheat honey MGO content is much lower than manuka honey. The pungent odor of buckwheat honey and its dark color may be mainly ascribed to its high mineral content. However, buckwheat honey holds phenolic compounds higher content than manuka honey [30].

The mechanism on how honey can promote an anticancer effect is of great interest. To this aim, we tested on A431, a cancer cell line, the cytotoxicity of three honey types characterized by different concentrations of polyphenolic compounds. We observed a growing cytotoxic effect as well from acacia < buckwheat < manuka honey, so we decided to use only manuka honey for the experiments and characterizations.

We have already tested the cytotoxicity of honeys on an epidermal non-cancerous cell line, i.e., a HaCaT cell line [2,29], which showed a lower toxicity than observed with the A431 cell line, especially highlighted by the EC_05_ values. On HaCaT cells, 24 h manuka honey treatment stimulated cells to allow a faster closure of the wound bed [29]. By contrast, in A431 cells, manuka honey determined the induction of apoptosis after only 3 h treatment.

Moving on to disclose the honey mechanism of action on tumor cells, we started evaluating the intracellular Ca^2+^ homeostasis by means of time-lapse confocal imaging. In the control condition, we did not observe any variation but following honey treatment we recorded a dose-dependent increase of the [Ca^2+^]_i_ followed by a sustained plateau trend only in the case of higher concentrations of the manuka honey treatment. In these conditions, unlike what happened after lower concentration exposure, the homeostatic [Ca^2+^]_i_ was not reached again, showing a behavior not compatible with cell survival.

To better characterize Ca^2+^ signaling occurring after manuka honey treatment, we decided to carry on a battery of subsequent experiments only with the concentration of 4% v/v, which was the most effective concentration compatible with the acquisition timing.

Ca^2+^ signaling in non-excitable cells includes the release of Ca^2+^ from intracellular stores and across the plasma membrane. The activation of the Ca^2+^ entry as well as the activation of Ca^2+^ pumps and the inhibition of passive Ca^2+^ pathways maintain homeostatic free cytosolic Ca^2+^ concentrations. In these kinds of cells, it is known that activation of PLC-mediated signaling pathways determines the release of Ca^2+^ from intracellular stores [31].

As cited above, the effect of honey on cells was carried out through the action of the H_2_O_2_ produced in the extracellular space [2]. Pretreatment with 500 U of catalase (CAT) completely abolished the Ca^2+^ rise, suggesting a pivotal role of ROS species to trigger the Ca^2+^ response. To unveil the origin of the increase of [Ca^2+^]_i_ after honey exposure, we performed again confocal observations treated with 4% manuka honey but in a 0 Ca^2+^ condition. The results showed that the entry from the extracellular space was fundamental for the observed Ca^2+^ peak; however, in the second part of the recording, we noticed a slight increase in the [Ca^2+^]_i_ compared with the basal one.

It has already been described that the initial Ca^2+^ rise could be caused by some ROS-sensitive Ca^2+^ channels such as TRPM2, TRPC5, TRPV1 and TRPA1 [32]; so, by using econazole, we observed a behavior similar to that recorded in 0 Ca^2+^, pointing out that a TRPM2 channel was involved in the initial Ca^2+^ entry from the extracellular space.

In many cell types, H_2_O_2_ can also stimulate the mobilization of Ca^2+^ by altering from the ER the Ca^2+^ release acting on ryanodine receptors [21] and inositol 1,4,5-trisphosphate (IP_3_)-dependent Ca^2+^ channels [33]. Moreover, it is known that H_2_O_2_ activates PLC in some cell types [34] so we explored whether the PLC/IP_3_ axis was activated after honey treatment. We characterized the involvement of the IP_3_ signaling by using a PLC inhibitor, U73122, and an IP_3_R inhibitor, caffeine, observing a significant but non-complete decrease of the [Ca^2+^]_i_ level.

AQP3 has been recognized as a mediator of H_2_O_2_ uptake and accumulation into the cytosol, which could modify the gating of adjacent H_2_O_2_-sensitive channels [13]. Functional experiments of water permeability demonstrated that the presence of 50 M H_2_O_2_ reduced significantly the AQP permeability, thus limiting its entry into the cell, as previously demonstrated in other cell models [12,35,36]. The concentration of H_2_O_2_ was similar to that obtained by manuka honey treatment [2] but, surprisingly, the treatment with manuka honey did not decrease the water permeability. This result may suggest that one or more components of manuka honey maintained the pore completely open even allowing the entry of high concentrations of H_2_O_2_ into the cells and thus leading to cell death even with high concentrations of H_2_O_2,_ which normally reduces its permeability.

We observed that the cytotoxic effect of manuka honey is carried out by apoptosis as already proposed [37] through the action of the H_2_O_2_, with a significant reduction of ΔψM and an increase of the intracellular ROS species while after pre-treatment with CAT, cell viability was preserved.

We also recently revealed in non-cancerous cells [2] that honey caused an overexpression of AQP3 and acted through this protein leading to an increase of intracellular ROS and triggering a variation in Ca^2+^ signaling. We demonstrated that also in the A431 cell line, honey determined an augmented expression of AQP3, which played a pivotal role in the carry-out of this mechanism even if it cannot be excluded that manuka honey may also act on other AQPs other than AQP3 [35,36]. Therefore, AQP3 facilitated H_2_O_2_ entry into the cytoplasm where it could trigger TRPM2-mediated Ca^2+^ entry from the extracellular space by acting from the cytosolic side. Moreover, as previously described by Huber and colleagues [38], the oxidative stress induced by H_2_O_2_ entry through AQP3, as well as the initial H_2_O_2_-mediated [Ca^2+^]_i_ rise, generated also a Ca^2+^ release from the ER.

Wang and co-workers [39] demonstrated a TRPM2–Ca^2+^–CaMKII–ROS signaling loop that could determine the shift between cell survival or death. They also pointed out that ROS produced or entered in the cytoplasm determined an activation of TRPM2 and a consequent Ca^2+^ influx, which in turn led to a further intracellular ROS production. Taking into account that intracellular ROS presence and/or the consumption of antioxidant proteins is normally augmented in tumor cells than in normal cells, making tumor cells more sensitive to oxidative stress [40], the proposed mechanism could be considered a positive feedback loop.

Evidence is growing that honey may have the potential to be an anticancer agent through several mechanisms. Here we described for the first time in a cancer cell line a possible mechanism through which honey could induce an alteration in the intracellular ROS and homeostatic balance of [Ca^2+^]_i_ leading to cell death by apoptosis. This mechanism seems to be enhanced by manuka honey’s ability to maintain a high H_2_O_2_ permeability. Thus, the potential anticancer activity of manuka honey may represent a novel tandem mechanism of a channel (AQP3) gating coupled with an H_2_O_2_-mediated apoptosis. Moreover, we also confirmed how an oxidative stress-induced disturbance passes through the alteration of intracellular Ca^2+^.

## 5. Conclusions

The confirmation of honey mechanisms of action through H_2_O_2_-AQP3-Ca^2+^ disturbance not only in a keratinocytes cell line as we have previously demonstrated [2] but also to a different extent in an A431 cell line does not diminish but rather encourages proof of the numerous anecdotal data about the health benefits and properties of honey. Taken together, data from in vitro experiments and preliminary in vivo researches are encouraging for using honey in chemo-prevention as well as an adjunct therapy to tumor drugs.

Moreover, honey, a most sustainable food produced naturally, is widely available, inexpensive and has minimal danger of adverse side effects [8]. However, there are still many unanswered questions before its use, i.e., the honey precise composition and properties and its antitumor characteristics may differ with the floral source, climate and honey bee species as well as geographic area and storage.

Therefore, further mechanistic studies (i.e., the involvement of mitochondria in the altered [Ca^2+^]_i_ buffering and apoptosis induction) as well as prospective randomized controlled clinical trials are a prerequisite to corroborate the antitumor potential of honey before endorsements for its use either alone or as an adjuvant therapy for neoplasm treatment.

## Figures and Tables

**Figure 1 life-10-00256-f001:**
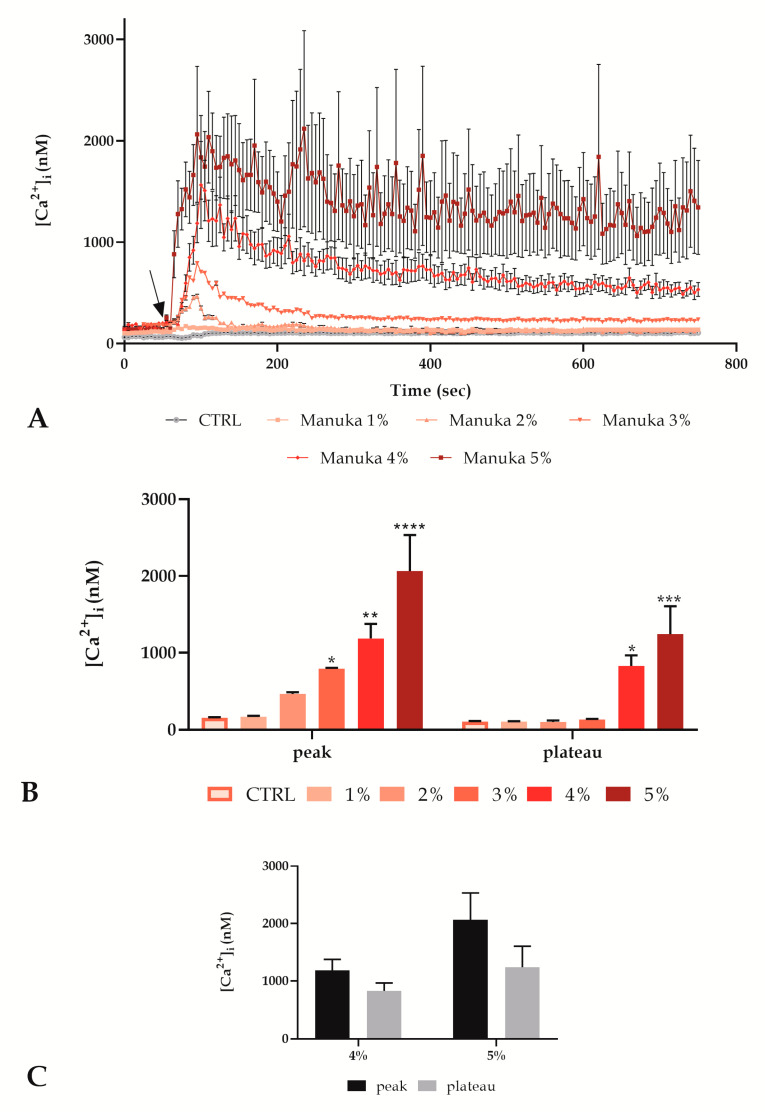
Manuka honey determined an intracellular Ca^2+^ concentration increase in A431 cells in a dose-dependent manner. (**A**) [Ca^2+^]_i_ variations assessed at 5 s intervals displaying no alterations in control conditions and dose-dependent configurations of Ca^2+^ signaling after treatment with different manuka honey concentrations, i.e., 1, 2, 3, 4 and 5% v/v. The arrow shows the honey addition after 60 s. Data are mean ± SEM of [Ca^2+^]_i_ traces assessed in different cells. Sample size: for each concentration, 40 cells from 3 exp. (**B**) [Ca^2+^]_i_ variations expressed as mean ± SEM of the peak or plateau of Ca^2+^ responses stimulated by treatment with different concentrations. Number of cells as previously indicated. Asterisks on bars indicate the statistical differences between CTRL and other conditions determined by two-way ANOVA followed by Tukey’s post-test (**** *p* < 0.0001, *** *p* < 0.001, ** *p* < 0.01, * *p* < 0.05). (**C**) Comparison of [Ca^2+^]_i_ variations expressed as mean ± SEM of the peak and plateau of Ca^2+^ responses stimulated by treatment with 4 or 5% v/v. Sample size as previous indicated. Statistics obtained by two-way ANOVA followed by Bonferroni correction.

**Figure 2 life-10-00256-f002:**
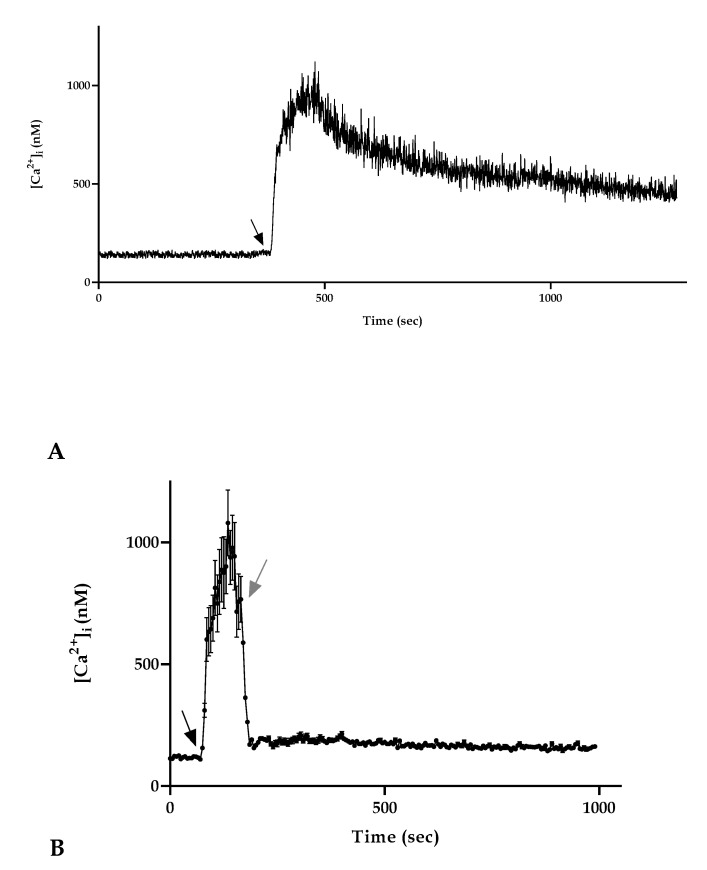
Characterization of the manuka honey induced Ca^2+^ increase in A431 cells. (**A**) [Ca^2+^]_i_ variations recorded at 0.5 s intervals induced by 4% manuka honey. Data are mean of [Ca^2+^]_i_ traces recorded in 40 different cells. (**B**) To monitor the role of honey in inducing a [Ca^2+^]_i_ increase, A431 cells were stimulated with 4% manuka honey (black arrow) and then the honey treatment was removed (gray arrow). When honey was removed from the medium, there was a [Ca^2+^]_i_ decrease. Data are mean ± SEM of [Ca^2+^]_i_ traces recorded in 40 different cells from 3 exp.

**Figure 3 life-10-00256-f003:**
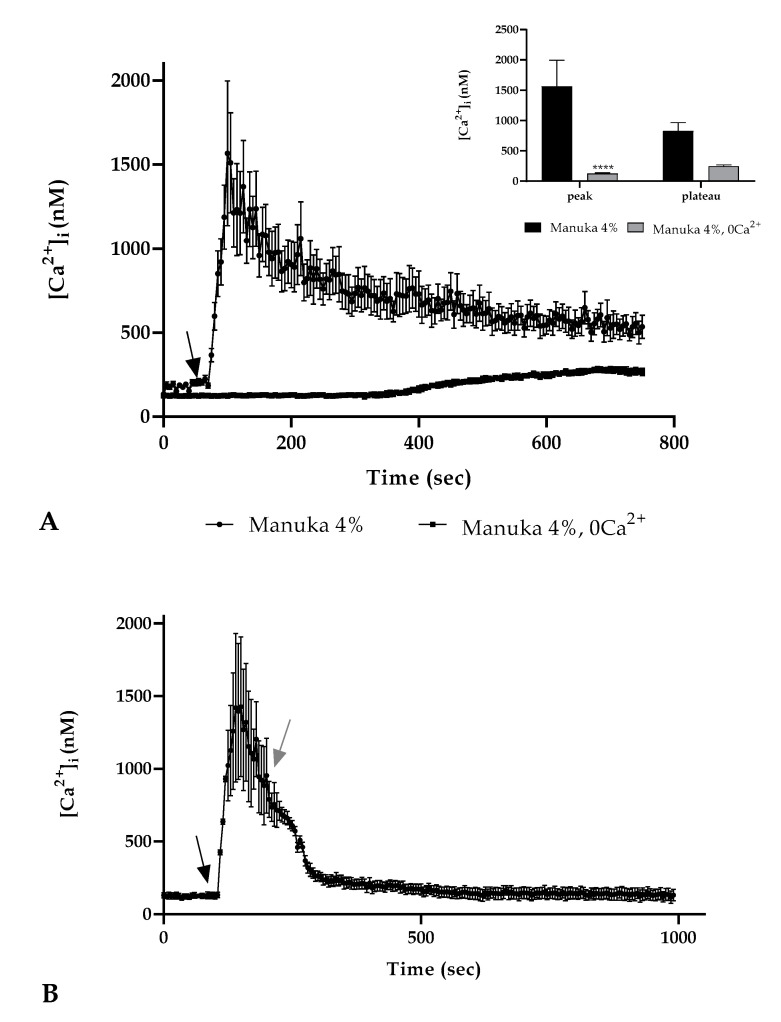
The Ca^2+^ response to honey involved extracellular Ca^2+^ entry. (**A**) The Ca^2+^ signaling due to 4% manuka honey exposure was eliminated in a 0 Ca^2+^ condition. The arrow shows the honey addition after 60 s. Data are expressed as mean ± SEM of [Ca^2+^]_i_ traces documented in different cells. Number of cells: manuka honey 0 Ca^2+^: 30 cells from 3 exp; manuka honey: 40 cells from 3 exp; **Insert.** Mean ± SEM of the peak Ca^2+^ response measured under the chosen treatments. Number of cells as in (**A**) Asterisks on bars indicate the statistical differences determined by two-way ANOVA followed by Bonferroni correction (**** *p* < 0.0001). (**B**) A431 cells have been stimulated with 4% manuka honey in the presence then in the absence of external Ca^2+^ (0 Ca^2+^). When Ca^2+^ was removed from the medium there was a [Ca^2+^]_i_ decrease. The black arrow indicates the addition of honey after 60 s; the gray arrow the removal of extracellular Ca^2+^. Data are expressed as mean ± SEM of [Ca^2+^]_i_ traces documented in 40 diverse cells from 3 exp.

**Figure 4 life-10-00256-f004:**
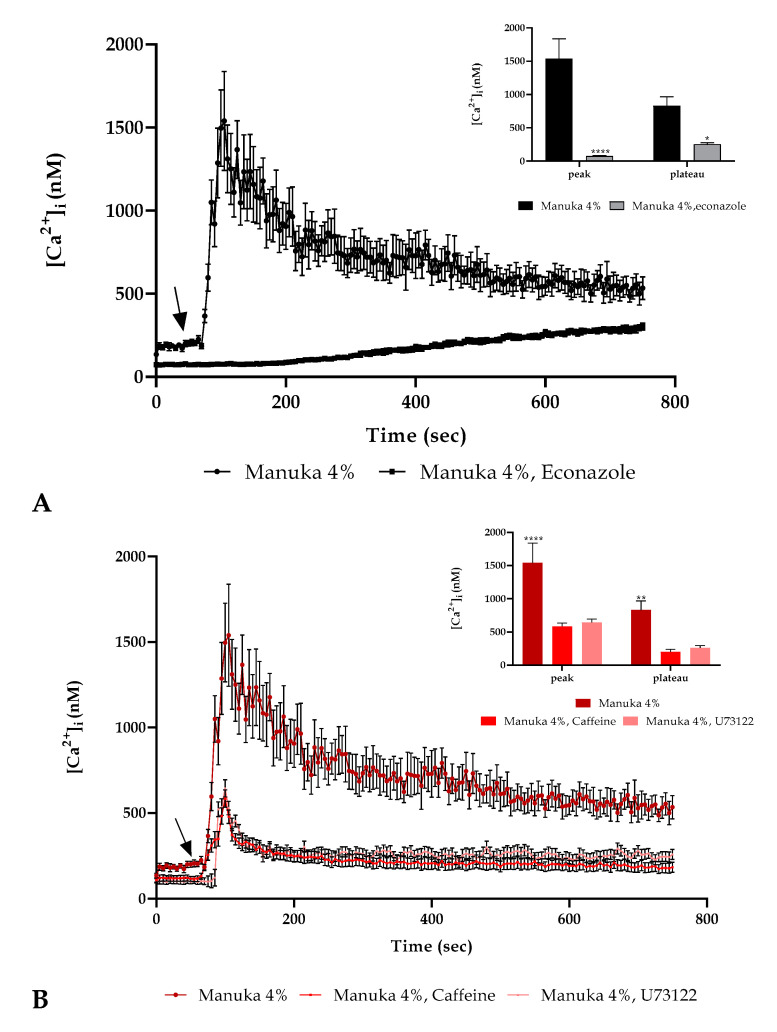
Ca^2+^ toolkit involvement. (**A**) The Ca^2+^ signaling response due to 4% manuka honey was significantly reduced in the presence of econazole (10 µM, 30 min pre-incubation). The arrow shows the honey addition after 60 s. Data are mean ± SEM of [Ca^2+^]_i_ traces assessed in different cells. Sample size: manuka honey + econazole: 40 cells from 3 exp; manuka honey: 40 cells from 3 exp. **Insert.** Mean ± SEM of the peak Ca^2+^ response measured under the chosen treatments. Number of cells as in A. Asterisks on bars indicate the statistical differences determined by two-way ANOVA followed by Bonferroni correction (**** *p* < 0.0001, * *p* < 0.05). (**B**) The Ca^2+^ response to 4% manuka honey was reduced in the presence of U73122 (10 µM, 30 min pre-incubation) and caffeine (10 mM, 30 min pre-incubation). The arrow specifies honey addition after 60 s. Data are mean ± SEM of [Ca^2+^]_i_ traces measured in cells. Number of cells: manuka honey: 40 cells from 3 exp; manuka honey + U73122: 40 cells from 3 exp; manuka honey + caffeine: 40 cells from 3 exp. **Insert.** Mean ± SEM of the peak Ca^2+^ response measured under the indicated treatments. Sample size as in B. Asterisks as in insert A (**** *p* < 0.0001, ** *p* < 0.01).

**Figure 5 life-10-00256-f005:**
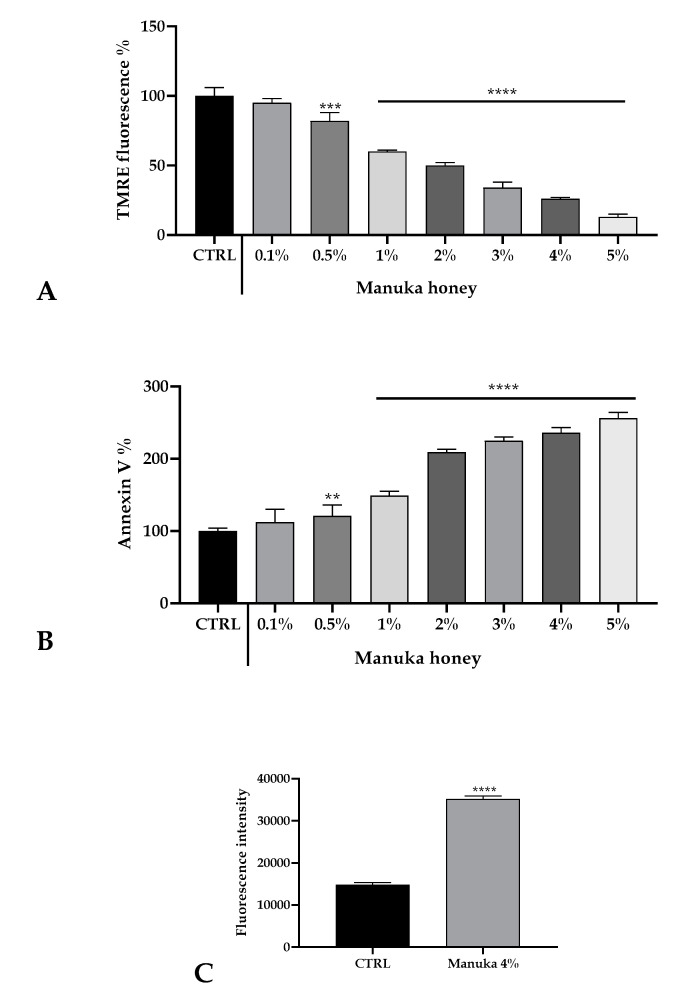
Honey induced cell death of A431 cells. (**A**) TMRE fluorescence evaluated in A431 cells treated for 3 h with varying concentrations of manuka honey. Data are expressed as mean ± SD obtained from 10 independent treatments and indicated as fluorescence %. Asterisks on bars indicate statistically significant differences assessed by one-way ANOVA followed by a Dunnet post-test (** *p* < 0.01, **** *p* < 0.0001). (**B**) Annexin V-FITC fluorescence in A431 cells treated with honey as above. Data are mean ± SD obtained from 10 independent treatments and expressed as fluorescence %. Statistics as in A (*** *p* < 0.001, **** *p* < 0.0001). (**C**). Fluorescence values recorded after 45 min incubation with 4% manuka honey. Data are indicated as mean ± SD of DHR-123 fluorescence measured in arbitrary units; n = 16 microplate wells from two experiments. Different asterisks on bars specify statistical differences determined by a *t*-test (**** *p* < 0.0001).

**Figure 6 life-10-00256-f006:**
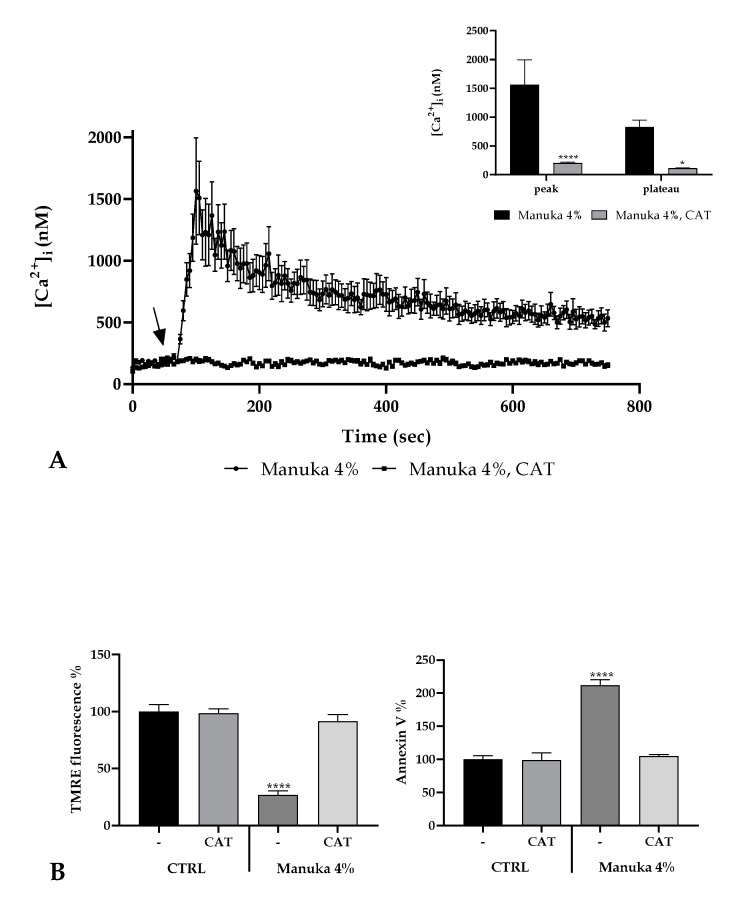
ROS involvement in the manuka honey mechanism of action. (**A**) The Ca^2+^ signaling due to 4% manuka honey was completely abrogated by treatment with Catalase (CAT 500U, 30 min pre-incubation). Data are indicated as mean ± SEM of [Ca^2+^]_i_ traces measured in different cells. The arrow shows manuka honey addition after 60 s. Sample size: manuka honey + CAT: 40 cells from 3 exp; manuka honey: 40 cells from 3 exp. **Insert**. Mean ± SEM of the Ca^2+^ peak response measured under the indicated honey exposures. Sample size as in A. Asterisks on bars specify statistical changes determined by Two-way ANOVA followed by Bonferroni correction (**** *p* < 0.0001, * *p* < 0.05). (**B**) TMRE fluorescence (left panel) and Annexin V-FITC fluorescence (right panel) measured in A431 cells treated with 4% manuka honey plus or not 500U CAT. Data are expressed as mean ± SD obtained from 10 independent experiments and expressed as fluorescence %. Asterisks on bars show statistically significant changes evaluated by one-way ANOVA followed by Bonferroni correction (**** *p* < 0.0001).

**Figure 7 life-10-00256-f007:**
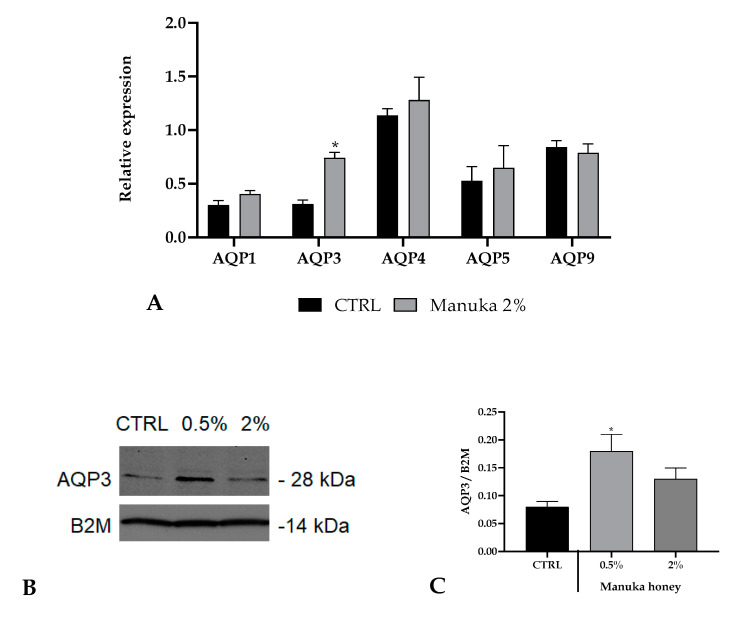
Expression of aquaporins (AQPs) in A431 cells. (**A**) Expression of AQP genes in A431 cells treated with 2% manuka honey. The mRNA level of AQPs was assessed by qRT-PCR and is expressed as mean relative expression ± SD (n = 3). Asterisk above the bar indicates statistical changes assessed by two-way ANOVA followed by Bonferroni correction (* *p* < 0.05). (**B**,**C**) Aquaporin-3 (AQP3) protein expression in A431 cells after manuka honey treatment (0.5 and 2%). CTRL is control condition. Blots illustrative of three were presented. Lanes were loaded with 30 μg of proteins, probed with an anti-AQP3 rabbit antibody as defined in the Materials and Methods section. The same blots were stripped and re-incubated with an anti-beta-2-microglobulin (B2M) antibody as housekeeping. A main band of about 28 kDa was shown for AQP3.

**Figure 8 life-10-00256-f008:**
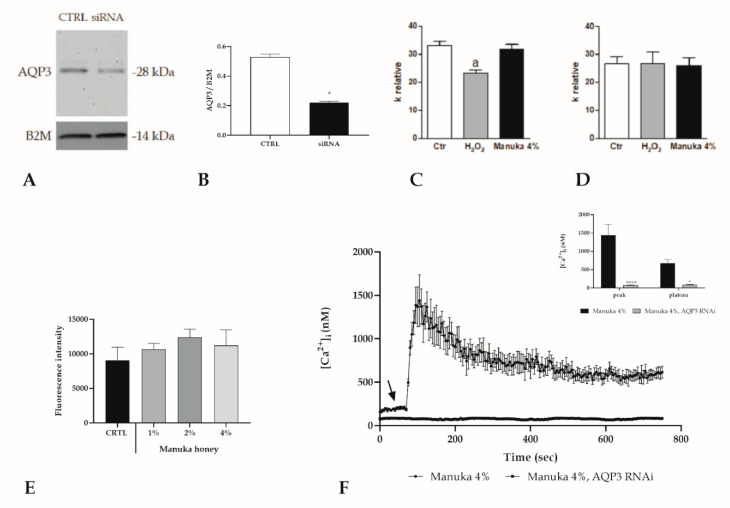
Pivotal role of AQP3 in mediating manuka honey cytotoxicity in A431 cells. (**A**,**B**) AQP3 protein levels in A431 cells in control conditions (CTRL) or after AQP3 RNAi (siRNA) treatment. Blots illustrative of three were presented. 30 μg of proteins were loaded for each lane then probed with an anti-AQP3 antibody and processed as specified in the Materials and Methods section. The same blots were then stripped and incubated with an antibody against anti-beta-2-microglobulin (B2M) as housekeeping (* *p* < 0.001, *t*-test). (**C**,**D**) Effect of H_2_O_2_ and 4% Manuka on the water osmotic permeability of A431 cells wild type and AQP3-KO. Cells were exposed to a 150 mOsm osmotic gradient in three different settings: untreated cells (Ctr), cells treated with 50 mM H_2_O_2_ for 45 min and treated with 4% manuka honey for 45 min. Bars indicate the osmotic water permeability of A431 cells expressed as a k relative. Values are expressed as mean ± SEM of 4–15 single shots for each of four different experiments. *p* < 0.05 vs. Ctr and Manuka 4% (ANOVA followed by a Newman–Keuls Q test). (**E**) Fluorescence values assessed at 10 min in silenced (RNAi AQP3) cells, incubated with increasing manuka honey concentrations (1, 2 and 4%). Data are shown as mean ± SD of rhodamine 123 fluorescence expressed in arbitrary units; n = 16 micro-plate wells from two different experiments. Statistics determined by one-way ANOVA followed by a Dunnet post-test. (**F**) The Ca^2+^ response to 4% v/v manuka honey was inhibited in A431 cells transfected with the RNAi selectively targeting AQP3. The arrow specifies the addition of honey after 60 s. Data are indicated as mean ± SEM of [Ca^2+^]_i_ traces recorded in different cells. Number of cells: manuka honey + siRNA: 40 cells from 3 exp; manuka honey: 40 cells from 3 exp. **Insert**. Mean ± SEM of the peak Ca^2+^ response recorded under the designated treatments. Number of cells as in D. Asterisks on bars indicate statistically different changes assessed by two-way ANOVA followed by Bonferroni correction (**** *p* < 0.0001, * *p* < 0.05).

**Table 1 life-10-00256-t001:** Primers sequences utilized for quantitative reverse transcriptase PCR (qRT-PCR).

Target Gene	Forward Sequences	Reverse Sequences
AQP9	5ʹ-ATTGGGATCCACTTCACTG-3ʹ	5ʹ-AGTGGACTGTGAACTTCC-3ʹ
AQP5	5ʹ-GCTGGCACTCTGCATCTTCGC-3ʹ	5ʹ-AGGTAGAAGTAAAGGATGGCAGC-3ʹ
AQP4	5ʹ-GCTGTGATTCCAAACGGACTGATC-3ʹ	5ʹ-CTGACTCCTGTTGTCCTCCACCTC-3ʹ
AQP3	5ʹ-CTGTGTATGTGTATGTCTGC-3ʹ	5ʹ-TTATGACCTGACTTCACTCC-3ʹ
AQP1	5ʹ-TAAGGAGAGGAAAGTTCCAG-3ʹ	5ʹ-AAAGGCAGACATACACATAC-3ʹ
β-actin	5ʹ-TCCCTGGAGAAGAGCTACGA-3ʹ	5ʹ-AGCACTGTGTTGGCGTACAG-3ʹ
GADPH	5ʹ-AATCCCATCACCATCTTCCA-3ʹ	5ʹ-TGGACTCCACGACGTACTCA-3ʹ

**Table 2 life-10-00256-t002:** siRNA oligonucleotide sequences.

Protein Target	Forward Sequence	Reverse Sequence
AQP3	5ʹ-GAGCAGAUCUGAGUGGGCA-3ʹ	5ʹ-UGCCCACUCAGAUCUGCUC-3ʹ

**Table 3 life-10-00256-t003:** Effective concentrations values, EC_05_ and EC_50_ (% v/v), resulting from dose-response curves for the three honeys on A431 cells at 24 h.

Honey	EC05	EC50
Acacia	2.36% (1.68–3.32%)	5.84% (5.33–6.39%)
Buckwheat	1.05% (0.67–1.69%)	3.78% (3.41–4.19%)
Manuka	0.66% (0.48–0.92%)	2.59% (2.36–2.85%)

Experiments were performed by the Calcein-Am method in triplicate with a minimum of eight replicates each. 95% CIs are indicated in parenthesis. Am, acetoxymethylester; CI, confidence interval; EC, effective concentration.

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
