# Peer review of "Manuka Honey Induces Apoptosis of Epithelial Cancer Cells through Aquaporin-3 and Calcium Signaling"

_life, 2020, doi:10.3390/life10110256_

Round 1
Reviewer 1 Report
I accept most of authors responses. However, most of responses should be integral part of the manuscript, for example:
- "The chemical composition of manuka honey is already described (Alvarez-Suarez et al., Foods 2014), with special attention given to its great polyphenolic composition and other bioactive compounds, such as methylglyoxal (MGO). MGO is a compound found in most types of honey, but usually only in small quantities. Our honey sample contains 250 mg/kg MGO (250+)".
- Authors claimed, that "Honey contains a huge amounts of phytochemicals such as high flavonoid and phenolic content which 373 contribute to its action [1]." is too low. Discussion section should be deeper, especially therefore, it is not sure (for reader) if composition of the strongest research honey is known or remains unknown. Moreover, authors never described the most important component of manuka honey (methyl gloxal). It is known, that methyl gloxal exhibited cytotoxic activity, therefore cytotoxicy of manuka honey is expected. Look here (for example):
https://www.cell.com/fulltext/S0092-8674(06)00050-X
https://www.ncbi.nlm.nih.gov/pmc/articles/PMC4206746/
Moreover, methylgloxal is expected to work with aquaporins:
"Fur and MGly at concentrations reported in traditional PDSs (Fur 0.8 microM; MGly 35 microM) significantly up-regulated eNOS mRNA and tended to down-regulate AQP1 mRNA in cultured endothelial cells."
https://pubmed.ncbi.nlm.nih.gov/19229826/
- Apart from methylgloxal, buckwheat honey is rich in active ketones. Therefore, authors exhibited its cytotoxicy discussion should mention about them.
- All used statistic test have to be described in material and method sections. Description of software is insufficient.
Minor issues.
Abstract: Please use full name instead shorts (especially [Ca2+]i and AQP3). It is better avoid shorts in abstracts.
Line 51. Please explain short AQP. It is used the first time in regular text.
Line 213. Please change ….0Ca2+… on 0 Ca2+
Author Response
I accept most of authors responses. However, most of responses should be integral part of the manuscript, for example:
We thank reviewer to taking the time to analyse our ms and to suggest some modifications.
- "The chemical composition of manuka honey is already described (Alvarez-Suarez et al., Foods 2014), with special attention given to its great polyphenolic composition and other bioactive compounds, such as methylglyoxal (MGO). MGO is a compound found in most types of honey, but usually only in small quantities. Our honey sample contains 250 mg/kg MGO (250+)".
We have modified the Materials and Methods section with the details of MGO content, as well as the Discussion section, improving the details about MGO presence and role.
- Authors claimed, that "Honey contains a huge amounts of phytochemicals such as high flavonoid and phenolic content which 373 contribute to its action [1]." is too low. Discussion section should be deeper, especially therefore, it is not sure (for reader) if composition of the strongest research honey is known or remains unknown. Moreover, authors never described the most important component of manuka honey (methyl gloxal). It is known, that methyl gloxal exhibited cytotoxic activity, therefore cytotoxicy of manuka honey is expected. Look here (for example):
https://www.cell.com/fulltext/S0092-8674(06)00050-X
https://www.ncbi.nlm.nih.gov/pmc/articles/PMC4206746/
We agree with the reviewer for the roles exerted by methylglyoxal (MGO). In fact, some studies are investigating methylglyoxal induction in mitochondrial dysfunction and cell death in liver. Some results indicated that oxidative stress is involved in methylglyoxal-induced toxicity, but methylglyoxal induced a significant but relatively small increase in ROS formation.
We know that methylglyoxal has been detected in a broad range of commercial food products and beverages, including bread, toast, tomatoes, boiled potatoes, caramelized sucrose, soya sauce and soya bean paste, roast turkey, alcohol from sugar cane, wine, saké, apple brandy and bourbon whiskey, apple, orange and tomato juices, maple syrup, beer, root-beer and cola, non-fat dry milk, instant, brewed and decaffeinated coffees, and cocoa and instant tea. No tumour was found that could be ascribed to administration of methylglyoxal (Fujita et al., 1986).
More in general, the 1,2-dicarbonyl compounds, including MGO, glyoxal and 3-deoxyglucosone, are generated either endogenously by cell metabolism, glucose oxidation and lipid peroxidation or by degradation of carbohydrates in foods and beverages. Highly reactive dicarbonyls attack the lysine, arginine and cysteine residues of long-lived proteins, such as collagens, to form irreversible AGEs causing changes in collagen pathophysiology that result in disruption of normal collagen matrix remodelling. So, some authors have suggested the concern that MGO in manuka honey may delay wound healing in diabetic patients. Further detailed researches are needed to fully elucidate the participation of honey/derived MGO in healing diabetic ulcers (Majtan, Evid Based Complem Altern Med 2011).
No other information about the eventual negative effects of MGO present in honey are available.
Recent work investigated methylglyoxal, a metabolite generated in cancer cells by enhanced aerobic glycolysis in so called “Warburg effect”, as a trigger of metastatic through MEK/ERK/SMAD1 pathway activation in breast cancer (Nokin et al, Breast Cancer Research 2019).
Moreover, methylgloxal is expected to work with aquaporins:
"Fur and MGly at concentrations reported in traditional PDSs (Fur 0.8 microM; MGly 35 microM) significantly up-regulated eNOS mRNA and tended to down-regulate AQP1 mRNA in cultured endothelial cells."
https://pubmed.ncbi.nlm.nih.gov/19229826/
We thank reviewer for this suggestion. We have considered the role of MGO, but we exclude its role in the observed effects. In fig. 7A, we did not observe any significant variation of AQP1. Moreover, we must consider that only for some AQPs (AQP3, -8 and -9) a permeability to H2O2 has been demonstrated, while other AQPs, like human AQP1 and AQP4, have a very low transport capacity if any (Medraño-Fernandez I., Bestetti S., Bertolotti M., Bienert G.P., Bottino C., Laforenza U., Rubartelli A., Sitia R. Stress regulates aquaporin-8 permeability to impact cell growth and survival. Antioxid. Redox Signal. 2016;24:1031–1044. doi: 10.1089/ars.2016.6636. Miller E.W., Dickinson B.C., Chang C.J. Aquaporin-3 mediates hydrogen peroxide uptake to regulate downstream intracellular signaling. Proc. Natl. Acad. Sci. USA. 2010;107:15681–15686. doi: 10.1073/pnas.1005776107. Hara-Chikuma M., Chikuma S., Sugiyama Y., Kabashima K., Verkman A.S., Inoue S., Miyachi Y. Chemokine-dependent T cell migration requires aquaporin-3-mediated hydrogen peroxide uptake. J. Exp. Med. 2012;209:1743–1752. doi: 10.1084/jem.20112398. Umberto Laforenza, Giorgia Pellavio, Anna Lisa Marchetti, Claudia Omes, Federica Todaro, and Giulia Gastaldi, Aquaporin-Mediated Water and Hydrogen Peroxide Transport Is Involved in Normal Human Spermatozoa Functioning, Int J Mol Sci. 2017 Jan; 18(1): 66).
Besides, Majtan et al (J Med Food 2014) demonstrated that MGO affect negatively the hydrogen peroxide accumulation in the manuka honey (through the inhibition of glucose oxidase).
Slavica Pavlovic-Djuranovic and coworkers demonstrated that AQP3 excluded methylglyoxal (Biochimica et Biophysica Acta 2006).
- Apart from methylgloxal, buckwheat honey is rich in active ketones. Therefore, authors exhibited its cytotoxicy discussion should mention about them.
We have modified the Discussion section with more details about honey composition.
- All used statistic test have to be described in material and method sections. Description of software is insufficient.
We have listed all the test and subsequently post-test or correction applied.
Minor issues.
Abstract: Please use full name instead shorts (especially [Ca2+]i and AQP3). It is better avoid shorts in abstracts.
We have modified the shorts in the abstract section
Line 51. Please explain short AQP. It is used the first time in regular text.
We have explained it.
Line 213. Please change ….0Ca2+… on 0 Ca2+
We have changed it.
Reviewer 2 Report
The manuscript by Martinotti et al still presents many orthographic and grammatical errors. As I suggested in the previous review, the manuscript should have been submitted to language editing but I guess the authors did not. Please correct the following errors:
Line 45: correct “inductions of apoptosis induction”.
Line 50: change “help” with “helps”.
Line 51: change “AQP” with “AQP3”.
Line 69: briefly describe the cell line.
Line 73: add a comma after “calcein-acetoxymethylester (Calcein-AM)”.
Line 88: change the comma after “140 NaCl” with a period.
Line 102: specify what “A23187” is.
Line 114: cancel the bracket.
Line 121: cancel “and” after “after washing” and add a comma.
Line 124: add a comma “To estimate the molecular weights of the bands”.
Line 140: the sentence “Cells were…measured in the fluorescence microplate reader” has no sense. The cells cannot be measured, fluorescence can. Change “measured” with “fluorescence was measured with microplate reader…”.
Line 149: cancel the bracket.
Line 147: list all the tests used for the statistical analysis. Indicating only the software “GraphPad Prism 8” is not sufficient in the methods section.
I do not see the adjustments on the manuscript I asked in my previous first 3 comments, despite the response of the authors:
“Please check and correct orthography and grammar errors throughout the manuscript.”
We have corrected the mistakes.
I still found errors throughout the manuscript.
“Fig. 1A: the control trace is not visible in the graph. Please make it clear.”
We have inserted the control trace.
Fig. 1A is the same of the previous submission.
“Legend fig 1B: the authors indicate that “Asterisks on bars indicate statistical differences”. Statistical differences compared with what condition? Please specify. Even though it is known to most that the Tukey method compares each experimental group against each control group, it is worth specifying it to facilitate a reader not familiar with this kind of statistical test.”
We have specified it in the figure legend.
I do not find any specification.
Author Response
The manuscript by Martinotti et al still presents many orthographic and grammatical errors. As I suggested in the previous review, the manuscript should have been submitted to language editing but I guess the authors did not. Please correct the following errors:
We thank reviewer to taking the time to analyse our ms and we apologize for errors and mistakes.
Line 45: correct “inductions of apoptosis induction”.
Line 50: change “help” with “helps”.
Line 51: change “AQP” with “AQP3”.
Line 69: briefly describe the cell line.
Line 73: add a comma after “calcein-acetoxymethylester (Calcein-AM)”.
Line 88: change the comma after “140 NaCl” with a period.
Line 102: specify what “A23187” is.
Line 114: cancel the bracket.
Line 121: cancel “and” after “after washing” and add a comma.
Line 124: add a comma “To estimate the molecular weights of the bands”.
Line 140: the sentence “Cells were…measured in the fluorescence microplate reader” has no sense. The cells cannot be measured, fluorescence can. Change “measured” with “fluorescence was measured with microplate reader…”.
Line 149: cancel the bracket.
We have modified all these errors.
Line 147: list all the tests used for the statistical analysis. Indicating only the software “GraphPad Prism 8” is not sufficient in the methods section.
We have listed all the test and subsequently post-test or correction applied.
I do not see the adjustments on the manuscript I asked in my previous first 3 comments, despite the response of the authors:
“Please check and correct orthography and grammar errors throughout the manuscript.”
We have corrected the mistakes.
I still found errors throughout the manuscript.
We regret for the inconvenience, now we have better taken care of the spelling and grammar of the text.
“Fig. 1A: the control trace is not visible in the graph. Please make it clear.”
We have inserted the control trace.
Fig. 1A is the same of the previous submission.
We apologize for the mistake. Now, we have uploaded a revised version of the figure with the CTRL trace.
“Legend fig 1B: the authors indicate that “Asterisks on bars indicate statistical differences”. Statistical differences compared with what condition? Please specify. Even though it is known to most that the Tukey method compares each experimental group against each control group, it is worth specifying it to facilitate a reader not familiar with this kind of statistical test.”
We have specified it in the figure legend.
I do not find any specification.
We apologize for the mistake. Now, we have insert the specification.
Round 2
Reviewer 1 Report
Manuscript was greatly improved. In my opinion, authors may try to perform similar experiments with honey fractions and (or) single isolated components. I suggest performing HPLC-MS/MS and GC-MS (afetr silylation or methylation) analyses of honey fractions next time.
I only detect one minor issues:
Line 431 ...even if it cannot be excluded that manuka honey may also 431 act on other AQPs other than AQP3... please add any citation or examples and citation.
Author Response
Manuscript was greatly improved. In my opinion, authors may try to perform similar experiments with honey fractions and (or) single isolated components. I suggest performing HPLC-MS/MS and GC-MS (afetr silylation or methylation) analyses of honey fractions next time.
We thank reviewer for suggestions that improved the ms and one of the next steps of the research will be to further characterize honeys.
I only detect one minor issues:
Line 431 ...even if it cannot be excluded that manuka honey may also 431 act on other AQPs other than AQP3... please add any citation or examples and citation.
We have inserted a citation.
Reviewer 2 Report
The authors did bring the requested corrections to the manuscript.
Author Response
The authors did bring the requested corrections to the manuscript.
We thank reviewer for the suggestions that improved the quality of the ms.
This manuscript is a resubmission of an earlier submission. The following is a list of the peer review reports and author responses from that submission.
Round 1
Reviewer 1 Report
Review
Authors investigate antitumor mechanism of manuka honey by different ways. Their research are valuable, however composition of analyzed samples remained unknown. This requirements is basic for todays research. For this reason, I suggest major revision of the manuscript. Honey composition is often unstable, authors should perform additional analysis. In my opinion at least LC-MS research of honey profile should be sufficient. Apart from this, article contains also another thing which have to be correct. Manuka honey often contains methylglyoxal as active components. If this components is present in honey authors should determine this.
Major issues:
- Please modify title.
- Material and method section
- How many samples of honey were investigated? Please specify.
- How exactly honey samples were prepared? What was the final concentration of honey?
- How “loading buffer” was used? Please explain.
- Please modify title.
- Line 58 – what is UMF15+? Unifloral honey? Please explain the short.
- Line 61 and 65 - what is DMEM? Please explain the short.
- Line 168 ….. is capable to produce alterations in [Ca2+]i…. How alterations? Concertation of Ca2+? This sentence is unclear.
- How authors determined kind of honey? Was any pollen analysis performed?
Statistical analysis
- Which statistical methods were used? Under the pictures authors described, that two-way ANOVA was used. Please explain what exactly was compared in material and methods sections.
Discussion
Authors should discus which component or components of honey may be responsible for observed activity.
Author Response
Authors investigate antitumor mechanism of manuka honey by different ways. Their research are valuable, however composition of analyzed samples remained unknown. This requirements is basic for todays research. For this reason, I suggest major revision of the manuscript. Honey composition is often unstable, authors should perform additional analysis. In my opinion at least LC-MS research of honey profile should be sufficient. Apart from this, article contains also another thing which have to be correct. Manuka honey often contains methylglyoxal as active components. If this components is present in honey authors should determine this.
We thank reviewer for taking time to comment our ms.
Manuka honey, a monofloral dark honey derived from the manuka tree (Leptospermum scoparium), has greatly attracted the attention of researchers for its biological properties, especially its antimicrobial capacities.
The chemical composition of manuka honey is already described (Alvarez-Suarez et al., Foods 2014), with special attention given to its great polyphenolic composition and other bioactive compounds, such as methylglyoxal (MGO). MGO is a compound found in most types of honey, but usually only in small quantities. Our honey sample contains 250 mg/kg MGO (250+).
Major issues:
-Please modify title.
We have modified it
- Material and method section
How many samples of honey were investigated? Please specify.
We investigated three samples of honey of different floral origin, such as acacia, buckwheat, and manuka, less than 1-year old.
How exactly honey samples were prepared? What was the final concentration of honey?
Raw honeys were stored at room temperature, with minimal light exposure. Stock honey solutions were prepared by dissolving in warmed cell culture medium or loading buffer for confocal microscopy and sterilized using a 0.22 μm filter. Fresh preparations of honeys were made before each experiment.
Final concentration of honeys depends on the experiment type.
How “loading buffer” was used? Please explain.
The loading buffer is a classic buffer for confocal microscopy experiment. As indicated in materials and methods section, it consisted of (mM) 10 glucose, 1 MgCl2, 2 CaCl2, 10 HEPES pH 7.4, 5 KCl, 140 NaCl,
Line 58 – what is UMF15+? Unifloral honey? Please explain the short.
UMF stands for “Unique Manuka Factor” and is a grading system developed by the UMF Honey Association in New Zealand.
Line 61 and 65 - what is DMEM? Please explain the short.
DMEM stands for Dulbecco's Modified Eagle's medium. DMEM is a widely used basal medium for supporting the growth of many different mammalian cells.
Line 168 …. is capable to produce alterations in [Ca2+]i…. How alterations? Concertation of Ca2+? This sentence is unclear.
Calcium plays a pivotal role in mediating many important biological functions. The intracellular calcium concentration is tightly regulated by a variety of systems and mechanisms. [Ca2+]i is the way to indicate cytosolic calcium concentration.
How authors determined kind of honey? Was any pollen analysis performed?
The quality has been previously assessed (Ranzato et al., WRR 2012) as well as certified by supplier.
Statistical analysis
Which statistical methods were used? Under the pictures authors described, that two-way ANOVA was used. Please explain what exactly was compared in material and methods sections.
We indicated in each Figure Legend which statistical tests has been utilized as well as the correction utilized, according to data and according to comparison requested.
Discussion
Authors should discus which component or components of honey may be responsible for observed activity.
In fact, we have demonstrated (Martinotti et al., IJMS 2019) that honey is able to produce H2O2 and a specific aquaporin (i.e. aquaporin-3) help the passive H2O2 diffusion across the biological membranes. The H2O2 mediated transport through AQP3 is of physiological importance for downstream cellular signalling pathways, such as the intracellular Ca2+ signals onset.
In the ms currently under revision, we investigated the effect of manuka honey on the growth of cancer cells, using an in vitro approach. Our findings provide mechanistic support for the apoptosis induction in cancer cells by honey treatment and further underline that reactive oxygen species (ROS) produced by honey treatment could diffuse through AQP3 across the plasma membrane deregulating intracellular Ca2+ signals leading to cell death.
Manuka honey not only modified the expression of AQP3 at gene and protein level, but more interestingly we demonstrated that AQP3 is involved in the entrance of H2O2 into the cells.
Thus, the potential anticancer activity of manuka honey may represent a novel tandem mechanism of a channel (AQP3) gating coupled with an apoptosis H2O2-mediated.
Reviewer 2 Report
Dear Editor and Authors,
here are my comments for the manuscript entitled “AQP3 and Ca2+ signaling: two partners in crime for manuka honey effects on A431 epithelial cancer cells” by Martinotti S. and colleagues. Actually, I have some major concerns about this work.
- Please check and correct orthography and grammar errors throughout the manuscript.
- Fig. 1A: the control trace is not visible in the graph. Please make it clear.
- Legend fig 1B: the authors indicate that “Asterisks on bars indicate statistical differences”. Statistical differences compared with what condition? Please specify. Even though it is known to most that the Tukey method compares each experimental group against each control group, it is worth specifying it to facilitate a reader not familiar with this kind of statistical test.
- Line 180: please insert a figure or a table showing the different cell viability to different manuka honey concentrations since the authors refer to a best survival rate of cells after exposure to 4% manuka honey with respect to 5% without showing any data.
- Statistical analysis: again, the authors do not specify the condition to which the statistical significance is referred. Moreover, why did the authors use different comparison tests in different experiments? Please justify it.
- Also, some results are presented as means ±S.E.M., while others as means ± S.D. Please, clarify this choice.
- Fig. 7: the authors show that AQP3 protein expression is increased by 0.5% manuka honey, while the 2% did not show any effect on the protein level. On the contrary, AQP3 mRNA levels in cells treated with 2% manuka honey are statistically higher than the control. Clarify the reason why there is this difference at protein level between these two different concentrations of honey. Did the authors test the effect of other manuka honey concentrations on AQP3 mRNA levels?
- Fig. 8: the data in fig. 8C are not exhaustive to connect the unchanged permeability after manuka honey treatment to AQP3 activity. The effect of the honey on permeability can be linked to another AQP expressed in this cell line, even though AQPs mRNA levels are not affected by the honey. The absence of an effect on mRNA does not exclude an effect of honey on the protein level or on the functionality of the other AQPs. The authors should demonstrate a direct link between manuka honey and AQP3 activity to claim that “This data supports the involvement of AQP3 in the entrance of H2O2 into the cells and that one or more substances present in the manuka honey were able to maintain the pore completely open even with high concentrations of H2O2” (lines 317-319).
- In fig. 3B, the authors show that after removing the honey, intracellular calcium concentration goes back to basal levels. This indicates that the cells are still alive! In the other Fluo-3 experiments, the authors did not wash out the honey and obviously calcium levels are not restored to physiological values, recording a permanent plateau. Is this plateau due to the effect of manuka honey on other important proteins implicated in calcium homeostasis, such as SERCA, PMCA and other Ca transporters? The authors did not test the honey effect on these proteins. Moreover, the experiments showing the absence of calcium response after 0Ca or econazole treatments do not have any positive control in order to show that the cells are actually responsive.
- Overall, it is not possible to establish an apoptosis H2O2-mediated effect of manuka honey based on the data presented, without performing a specific apoptotic test.
Author Response
Dear Editor and Authors,
here are my comments for the manuscript entitled “AQP3 and Ca2+ signaling: two partners in crime for manuka honey effects on A431 epithelial cancer cells” by Martinotti S. and colleagues. Actually, I have some major concerns about this work.
We thank the reviewer for the possibility to clarify some points of our ms as well as the data and their interpretation.
Please check and correct orthography and grammar errors throughout the manuscript.
We have corrected the mistakes
Fig. 1A: the control trace is not visible in the graph. Please make it clear.
We have inserted the control trace.
Legend fig 1B: the authors indicate that “Asterisks on bars indicate statistical differences”. Statistical differences compared with what condition? Please specify. Even though it is known to most that the Tukey method compares each experimental group against each control group, it is worth specifying it to facilitate a reader not familiar with this kind of statistical test.
We have specified it in the figure legend.
Line 180: please insert a figure or a table showing the different cell viability to different manuka honey concentrations since the authors refer to a best survival rate of cells after exposure to 4% manuka honey with respect to 5% without showing any data.
Due to our short incubation period for confocal microscopy assessment of intracellular calcium concentration (1000 seconds), we cannot provide a consistent cell viability value comparable with the shortness of timing. The data on which we rely to highlight the highest toxicity of the 5% manuka honey are those obtained from confocal microscopy observation, where it is easy to observe the inability of the 5% recorded trace to return to homeostatic intracellular calcium values.
Statistical analysis: again, the authors do not specify the condition to which the statistical significance is referred. Moreover, why did the authors use different comparison tests in different experiments? Please justify it.
We decided to utilized the best suitable statistical test in any figures according to data and according to comparison requested. We indicated in each Figure Legend which statistical tests has been utilized as well as the correction utilized.
Also, some results are presented as means ±S.E.M., while others as means ± S.D. Please, clarify this choice.
The standard deviation (SD) measures the amount of variability, or dispersion, from the individual data values to the mean, while the standard error of the mean (SEM) measures how far the sample mean of the data is likely to be from the true population mean. The SEM gives an idea of the accuracy of the mean, and the SD gives an idea of the variability of single observations. For confocal data, the SEM quantifies how precisely we know the true mean of the population, taking into account both the value of the SD and the sample size.
Fig. 7: the authors show that AQP3 protein expression is increased by 0.5% manuka honey, while the 2% did not show any effect on the protein level. On the contrary, AQP3 mRNA levels in cells treated with 2% manuka honey are statistically higher than the control. Clarify the reason why there is this difference at protein level between these two different concentrations of honey. Did the authors test the effect of other manuka honey concentrations on AQP3 mRNA levels?
In order to understand the expression of which aquaporins were affected after manuka honey exposure, we decided to use the highest concentration that had no toxic effects. Once we observed that the expression of AQP3 was affected at mRNA level, we investigated at protein level using even a lower concentration more compatible with prolonged exposure to honey.
The discrepancy between the effect given by the 2% exposure at the mRNA and protein level, could be due to an early harmful effect, inducing a possible alteration in protein folding and therefore, despite the mRNA level present, a less protein content detectable.
Fig. 8: the data in fig. 8C are not exhaustive to connect the unchanged permeability after manuka honey treatment to AQP3 activity. The effect of the honey on permeability can be linked to another AQP expressed in this cell line, even though AQPs mRNA levels are not affected by the honey. The absence of an effect on mRNA does not exclude an effect of honey on the protein level or on the functionality of the other AQPs. The authors should demonstrate a direct link between manuka honey and AQP3 activity to claim that “This data supports the involvement of AQP3 in the entrance of H2O2 into the cells and that one or more substances present in the manuka honey were able to maintain the pore completely open even with high concentrations of H2O2” (lines 317-319).
We thank the reviewer for the opportunity to clarify this point.
We have evaluated again the water osmotic permeability of A431 cells wild type and AQP3-KO. Cells were exposed to a 150 mOsm osmotic gradient in three different conditions: untreated cells (Ctr), cells treated for 45 min with 50 mM H2O2 and treated for 45 min with 4% Manuka. So we have inserted a new panel (D) in figure 8. However, we cannot excluded that manuka may also act on other AQPs other than AQP3.
In fig. 3B, the authors show that after removing the honey, intracellular calcium concentration goes back to basal levels. This indicates that the cells are still alive! In the other Fluo-3 experiments, the authors did not wash out the honey and obviously calcium levels are not restored to physiological values, recording a permanent plateau. Is this plateau due to the effect of manuka honey on other important proteins implicated in calcium homeostasis, such as SERCA, PMCA and other Ca transporters? The authors did not test the honey effect on these proteins. Moreover, the experiments showing the absence of calcium response after 0Ca or econazole treatments do not have any positive control in order to show that the cells are actually responsive.
We thank the reviewer for the questions and for the possibility of clarifying these aspects. In fact, for figure 3B, we have performed the experiment starting with confocal microscopy buffer containing Ca2+, then after 60 seconds we have stimulated with 4% manuka honey (first arrow). After the peak, we have removed solution (containing Ca2+ ions). In this way we replaced the buffer with the equivalent buffer for confocal microscopy without Ca2+ ions, but maintaining 4% manuka honey treatment. So, when Ca2+ has been removed from the medium, despite the presence of 4% manuka honey, there was [Ca2+]i decrease.
Econazole, a TRPM2 inhibitor, has been utilized to correctly inhibit the entry of Ca2+ ions abrogating the Ca2+ peak after 4% manuka honey exposure. We have previous experience that TRPM2 did not affect other channels or cell viability as well as we tested (data not shown) that Ca2+ homeostasis is not affected by econazole treatment.
Overall, it is not possible to establish an apoptosis H2O2-mediated effect of manuka honey based on the data presented, without performing a specific apoptotic test.
We agree with the Reviewer that our ms is not directly aimed at evaluating honey and apoptosis molecular mechanism. However, with this study we explored the effects of manuka honey on A431 cancer cell lines, and in particular the relationship established between H2O2 honey-dependent production, AQP3-mediated transport and intracellular Ca2+ dysregulation.
We believe that, taken together, our data offer a convincing picture of the potentiality of manuka honey, even if we are absolutely aware that further studies are needed to show a complete characterization of their effects, in particular in term of specific apoptotic pathway activation after Ca2+ homeostasis alteration induced by H2O2 entry.
Taken together, data from in vitro experiments and preliminary in vivo researches are encouraging for honey in terms of chemo-prevention, as well as an adjunct therapy to cancer drugs. However, there are still many unanswered questions before its use, i.e. the honey precise composition and properties and its antitumor characteristic may vary with the floral source, climate, honey bee species, as well as geographical area, and storage.